# Precise timing of MIS 7 sub-stages from the Austrian Alps

Kathleen A. Wendt[a*], Xianglei Li[b], R. Lawrence Edwards[b], Hai Cheng[c,b], Christoph Spötl[a]

[a]Institute of Geology, University of Innsbruck, Innrain 52, 6020 Innsbruck, Austria

5  [b]Department of Earth Sciences, University of Minnesota, 116 Church Street SE, Minneapolis 55455, USA

[c]Institute of Global Environmental Change, Xi'an Jiaotong University, Xi'an 710049, China

*current address: College of Earth, Ocean, and Atmospheric Sciences, Oregon State University, Corvallis, OR 97331, USA; kathleen.wendt@oregonstate.edu

**Abstract.** Investigating the precise timing of regional-scale climate changes during glacial terminations and the interglacial periods that follow is key to unraveling the mechanisms behind these global climate shifts. Here, we present a high precision time series of climate changes in the Austrian Alps that coincide with the later portion of Termination III (TIII), the entire penultimate interglacial (Marine Isotope Stage (MIS) 7), Termination IIIa (TIIIa), 15  and the penultimate glacial inception (MIS 7/6 transition). Using state-of-the-art mass spectrometry techniques, we have constructed a uranium-series chronology with relative age uncertainties averaging 1.7‰ (2σ) for our study period (247 to 191 thousand years before present [ka]). Results reveal the onset of warming in the Austrian Alps associated with TIII at 242.5 ±0.2 ka and the duration of MIS 7e warming between 241.8 to 236.7 (±0.6) ka. An abrupt shift towards higher $\delta^{18}O$ values at 216.8 ka marks the onset of regional warming associated with TIIIa. Two 20  periods of high $\delta^{18}O$ values (greater than -10‰ VPDB) between 215.9–213.3 and 204.3–197.5 (±0.4) ka coincide with interglacial substages MIS 7c and 7a, respectively. Multiple fluorescent inclusions suggest a partial retreat of the local Alpine glacier during peak obliquity forcings at 214.3 ±0.4 ka. Two newly collected stalagmites from Spannagel Cave (SPA146 & 183) provide high-resolution replications of the latter portion of the MIS 7a/6e transition. The resulting multi-stalagmite record reveals important chronological constraints on climate shifts in the 25  Austrian Alps associated with MIS 7, while offering new insight into the timing of millennial-scale changes in the North Atlantic realm leading up to TIII and TIIIa.

# 1 Introduction

Marine Isotope Stage (MIS) 7 (ca. 246-186 thousand years before present [ka], where present is 1950 CE) stands distinctively apart from the last several interglacial cycles. Following the low-amplitude glacial Termination III (TIII), MIS 7e was a relatively weak interglacial that returned to glacial conditions (MIS 7d) within 20 thousand years (ky) (PAGES, 2016). Glacial conditions terminated with a second deglaciation (TIIIa), which gave way to a second interglacial with three distinct sub-stages (MIS 7c, b, a). Although the inception of an interglacial is ultimately paced by astronomical forcing, the precise timing (millennial-scale) of each glacial termination and the strength of the successive interglacial depends on various global climate parameters and feedbacks. Investigating these internal forcings requires the intercomparison of precisely dated climate records in order to identify leads, lags, and synchronicities across different climate zones. The role of forcings that led to the last two glacial terminations and the variability within the last two interglacial periods (MIS 5e and Holocene) have been well studied. Obtaining records from MIS 7 and the penultimate glaciation that contain sufficient resolution and chronological precision to make meaningful regional comparisons, however, has proved challenging. In the North Atlantic realm, marine records show that major meltwater pulse events punctuated a period of rapid sea surface warming associated with TIII and TIIIa (e.g. Channell et al., 2012; Hodell et al., 2008; Martrat et al., 2004, 2007). The chronologies of these marine records, however, are frequently dependent on orbital tuning or alignment to global benthic $\delta^{18}O$ stacks, which feature large uncertainties of $\pm10$ ka during MIS 7 (e.g. LR04 stack; Lisiecki and Raymo, 2005). In continental Europe, cave and lacustrine records indicate abrupt environmental shifts associated with the MIS 7 sub-stages (e.g. Tzedakis et al., 2004; Despart et al., 2006; Roucoux et al., 2008) and TIII (Pérez-Mejías et al., 2017) accompanied by regional temperature and atmospheric circulation changes (e.g. Spötl et al., 2008; Badertscher et al., 2011; Columbu et al., 2019) often in-step with North Atlantic climate change (Denniston et al., 2017). However very few terrestrial records span the entirety of MIS 7 and its glacial transitions. A lack of absolute-dated, continuous records hinders the ability to capture the timing and full duration of regional climate changes, resulting in critical gaps in knowledge.

This study aims to determine the onset of warming in continental Europe associated with TIII and TIIIa, as well as the regional climate variations during the five sub-stages of MIS 7. To do so, we turn to the Austrian Alps (Fig. 1). Over the last century, the Alps experienced twice the amplitude of temperature change relative to the mean Northern Hemisphere (Auer et al., 2007), contributing to a 45% reduction of glacier surface and roughly 50% of ice volume between 1900 and 2011 (Huss, 2012). This high degree of sensitivity renders the Austrian Alps an ideal location

to pinpoint the timing of continental temperature variations. Here, we focus on Spannagel cave — a high altitude marble cave located in the central Austrian Alps. Drip waters in this cave are fed by glacial and snowpack melt (Mangini et al., 2005). Past studies demonstrate that Spannagel speleothem $\delta^{18}O$ represents a robust proxy of local
winter temperatures and moisture sources, the latter being dominated by the North Atlantic Ocean (Mangini et al., 2005; see section 2.1). Pioneering work by Holzkämper et al. (2005) and Spötl et al. (2007, 2008) presented multiple MIS 7 stalagmites and flowstones deposited in Spannagel cave. Since these studies, new developments in the measurement precision of U and Th isotopes by Cheng et al. (2013) and Craig et al. (2016, 2017) allow for ultra-high age precision, with $2\sigma$ absolute age uncertainties averaging $\pm300$ years for MIS 7. In addition, two newly
discovered stalagmites from Spannagel Cave provide further insight into the penultimate glacial inception (MIS 7/6 transition). The resulting Spannagel $\delta^{18}O$ record provides a high-precision chronology of abrupt climate change in Europe during MIS 7, while addressing the need for absolute-dated paleorecords from regions that are sensitive to the North Atlantic realm.

## 2 Study site

Spannagel Cave (Fig. 1; 47°04'54"N 11°40'2"E; 2300 to 2530 m a.s.l.) is located above the timberline in the vicinity of a retreating glacier (Hintertux Glacier) close to the main Alpine crest. Previous publications provide details on the setting of this cave (Spötl et al., 2004; Spötl and Mangini 2007, 2010). A few salient features relevant for this study are summarized here. Spannagel cave developed in a ca. 20 m thick Upper Jurassic marble (Hochstegen Formation) sandwiched between gneiss bedrock. An interesting feature of Spannagel cave drip waters is that they
are rich in U. Cave water U concentrations range from 5 ppb (small cave stream) to 33 ppb (stalactite drip water). As a result, calcitic speleothems from this cave are exceptionally high in U concentrations (3 to 399 ppm). The U is most likely sourced from the overlying gneiss which yield significantly higher U concentrations (3.4–12.0 ppm) than the marble (0.3–2.2 ppm).

Spannagel is a well-ventilated cave system with stable air temperatures in the interior of the cave that stay within a narrow range of +1.8 to +2.2°C (measured at several sites using data loggers over a period of about 20 years). These values are about 2°C higher than the mean annual air temperature at the elevation of the cave, which is a reflection of the small positive thermal anomaly due to the ascending geometry of the cave (chimney effect; Spötl and Pavuza, 2016). Relative humidity in the cave interior is invariably higher than 96%. Today, the actively
retreating Hintertux Glacier terminates ca. 500 m south of Spannagel cave. During the last glacial maximum, the

cave was completed covered by as much as 150–250 m of glacial ice (van Husen, 1987). The deposition of speleothems during several cold climate periods suggests that the Hintertux Glacier has remained largely temperate, or warm-based (Spötl and Mangini, 2007). Under glacial conditions, it is thought that the oxidation of pyrite in the host rock allowed karst dissolution beneath the glacier without the input of soil-derived carbon dioxide (Spötl and Mangini, 2007).

Stalagmite SPA121 (Fig. S1) was found in the northern segment of the cave embedded in unconsolidated silty sand which was most likely transported into the cave by high-discharge streams during the last deglaciation. The stalagmite is 19.3 cm tall and was found attached to a platy and angular piece of calcite-cemented gravel. The surface of the sample is covered by a thin layer of medium gray, clay-rich calcite, but the interior preserves mostly transparent, inclusion-poor calcite, showing a striking pattern of cracks. The origin of these cracks is enigmatic but may be related to freezing of the cave or the sediments in which the sample was embedded during peak glacial periods. SPA183 is a 39 cm tall stalagmite collected from the central part of the cave and a slightly higher altitude than SPA 121. SPA183 revealed three distinct growth axes that are marked by variations in color (Fig. S1). SPA146 is a 24 cm tall stalagmite which grew in the same small chamber as SPA183. Both stalagmites were detached from their growth substrate and embedded in sand, but likely transported less than a few meters. SPA146 shows two distinct growth axes (Fig. S1).

## 2.1 Climate setting

The Austrian Alps are intermediately situated between the Westerlies and northerly flowing Mediterranean air masses. Back-trajectory studies show that regional precipitation is predominantly sourced from North Atlantic and Arctic Oceans (~60%) and the Mediterranean Sea (~20%) (Kaiser et al., 2002; Sodemann and Zubler, 2010). Seasonally, strong westerly flow during winter months transports moisture from the North Atlantic Ocean, whereas infrequent northerly flow from the Mediterranean Sea is more common during summer months (Kaiser et al., 2002; Sodemann and Zubler, 2010). Winter precipitation is stored in Alpine glaciers and snow cover. Glacial and snowpack melt is the dominant source of groundwater recharge in the Alps and the largest supplier to Spannagel dripwaters (Mangini et al., 2005; Spötl and Mangini, 2007). Variations in the $\delta^{18}O$ of calcite ($\delta^{18}O_c$) that precipitates from dripwaters have been shown to reflect variations in winter precipitation $\delta^{18}O$ (Mangini et al., 2005; Spötl et al., 2006; Spötl and Mangini, 2002, 2007).

The Austrian Network for Isotopes in Precipitation (ANIP) reveals a tight coupling between the mean annual $\delta^{18}O$ of regional precipitation and air temperatures (Kaiser et al., 2002) in concordance with other locations across the European Alps (e.g. Schürch et al., 2003; Field, 2010). On sub-annual scales, the mean monthly $\delta^{18}O$ of ANIP
stations show an almost parallel evolution with average air temperatures (Hager and Foelsche, 2015). Additional influences on $\delta^{18}O$ may include changes in the proportion of seasonal precipitation, which is linked to moisture source. Stable isotope analysis of rainfall collected from the Patscherkofel mountain station (20 km northwest of Hintertux; 2200 m a.s.l.) suggest that $\delta^{18}O$ precipitation sourced from the North Atlantic Ocean is depleted relative to Mediterranean sources, due to (i) the lower isotopic value of Atlantic Ocean water and (ii) a longer transport
pathway (Kaiser et al., 2002). However, due to the high elevation of our study site, orographic effects are likely to smooth any large source-based differences in precipitation $\delta^{18}O$.

Spannagel $\delta^{18}O_c$ is a robust proxy for (predominantly winter) surface air temperatures on annual to orbital timescales (Mangini et al., 2005; Spötl et al., 2006; Spötl and Mangini, 2002, 2007), as similarly documented in
speleothem records across the European Alps (e.g. Boch et al., 2011; Moseley et al., 2014; Johnston et al., 2018; Wilcox et al., 2020). Major shifts in Spannagel $\delta^{18}O_c$ may be further amplified by (i) changes to the $\delta^{18}O$ of moisture sources, such as meltwater pulses in the North Atlantic (e.g. Mangini et al., 2007), (ii) changes in the degree of moisture recycling along flow paths, and (iii) changes in the seasonal proportions of annual totals. For example, a northerly displaced polar front and warmer SSTs likely contributed to increased advection across the Mediterranean
during interglacial periods (Drysdale et al., 2009), resulting in a greater input of heavier $\delta^{18}O$ Mediterranean moisture to the Austrian Alps during the summer (Moseley et al., 2015). Such a change in atmospheric circulation would act as a positive, yet relatively minor, feedback to the temperature-dominated Spannagel $\delta^{18}O_c$ signal.

## 3 Methods

Stalagmites SPA121, 146, and 183 were halved and polished. Subsamples for U-Th dating (n=40) were hand drilled along the growth axis of halved stalagmites. Subsample trenches were drilled no larger than 1 mm in width, such that the sampling error (~175 years) remained within the analytical uncertainties (see Results). Target sample weights ranged from 200 to 20 mg in concordance with changes in U concentration. Subsamples were spiked with

a mixed $^{233}$U-$^{236}$U-$^{229}$Th spike similar to that described in Edwards et al. (1987). Procedures for U and Th chemical separation and preparation of reagent solutions follow the methods described in Edwards et al. (1987) and Shen et al. (2002).

U and Th isotopic measurements were made on a Thermo Scientific Neptune Plus MC-ICP-MS following the instrument calibration and the Faraday cup measurement method described in Cheng et al. (2013). In addition, a $10^{13}$ Ohm amplifier was installed within the detection system in order to collect low ion beam intensities (e.g. $^{234}$U and $^{230}$Th) (Craig et al., 2016, 2017 and references therein). The methods of gain calibration and dynamic time correction of the high resistor are largely based on Craig et al. (2016) and (2017). Each sample was measured for 300s or longer. Intensities of $^{234}$U and $^{230}$Th beams were on average 15 and 5 mV, respectively.

Stable isotope samples were micromilled continuously along the growth axis of each stalagmite at 0.15–0.20 mm increments. 946 stable isotope measurements on stalagmite SPA121 were previously published in Spötl et al. (2008). 890 and 583 new stable isotope measurements from stalagmites SPA183 and SPA146, respectively, are reported here. A total of 13 Hendy tests were drilled along individual growth layers from stalagmites SPA146 and SPA 183 (Figs. S4-5). All calcite powders were analyzed using a Gasbench II coupled with a Delta V Plus isotope ratio mass spectrometer. The 1-sigma precision is 0.06 and 0.08 ‰ for $\delta^{13}$C and $\delta^{18}$O, respectively. Results are reported relative to VPDB.

## 4 Results

Resulting U-Th ages and their respective replicates are in stratigraphic order within uncertainties (Table S1). This study focuses on the MIS 7 portion of stalagmite SPA121, which was deposited without interruption between 248.5 and 191.5 ($\pm$1) ka (see Spötl et al., 2008 for details on all SPA121 growth phases). New U-Th ages for this stalagmite fall within age uncertainties of the previously published ages (Spötl et al. 2008; Fig. 2). Similar $\delta^{234}$U$_i$ values and $^{232}$Th and $^{238}$U concentrations further underscore a high degree of reproducibility between the two SPA121 data sets, which were measured in different laboratories using different instruments (MC-ICP-MS vs. TIMS). New SPA121 ages improve the precision of previously published age uncertainties by an order of magnitude (Fig. 2), from an average of 1.7% to 0.17%.

The late MIS 7 growth phase of stalagmites SPA183 and SPA146 occurred between 191.9-190.6 (±0.6) ka and 191.6-182.3 (±0.3) ka, respectively. The exact onset of growth is unknown, as both stalagmites show evidence of diagenesis spanning first 2-3 cm of the late MIS 7 growth phase (Fig. S2; Table S1). Evidence of diagenetic

alteration includes a conspicuously white, milky calcite fabric and U-Th ages that are out of stratigraphic order and unable to be replicated. As a result, this study focuses only on the unaltered portion of the late MIS 7 growth phases of stalagmites SPA183 and SPA146 (Fig. S1). Relative age uncertainties of SPA146 and 183 average 0.09% and 0.16%, respectively. Significantly higher $\delta^{234}U_i$ values from SPA146 and 183 relative to SPA121 suggest differing drip sources between the first two neighboring stalagmites and SPA121. Growth rates calculated from new SPA121

ages closely agree with previously published data in Spötl et al. (2008) (Fig. 2). The average growth rate is 5.6 µm/yr, excluding one period of exceptionally low growth rate (0.8 µm/yr) between 231.1 and 219.6 (±0.6) ka. In contrast, average growth rates of SPA183 and SPA146 are higher (63 and 140 µm/yr, respectively). Differences in growth rates are likely due to differing drip sources.

SPA121 stable isotope values used in this study are from Spötl et al. (2008). New stable isotope data from SPA146 and 183 is reported in Table S2. The range of SPA121 $\delta^{18}O_c$ values (-8.1 to -14.7‰) and $\delta^{13}C$ values (9.7 to 0.8‰) is in agreement with newly measured $\delta^{18}O_c$ values from SPA146 and SPA183 (Fig. S3). Slight (<1‰) offsets in absolute values between stalagmites are observed, but do not influence the relative variations in stable isotopes which are the focus of this study. Hendy test results (Figs. S4–5; Spötl et al., 2008) show no evidence for significant

kinetic fractionation during the deposition of all three stalagmites.

The $\delta^{18}O_c$ signature of SPA stalagmites is depleted during the MIS 7 warm intervals (average -9.2‰) relative to the Holocene (average -7.8‰; Spötl et al., 2004), which suggests cooler winter temperatures during MIS 7. This is consistent with globally distributed evidence suggesting cooler Northern Hemisphere temperatures throughout MIS

7 (PAGES, 2016) in conjunction with lower sea levels (Robinson et al., 2002; Thompson and Goldstein, 2005; Dutton et al., 2009; Andersen et al., 2010; Murray-Wallace, 2002) and atmospheric $p$CO_2 (Bazin et al., 2013) relative to the Holocene.

The $\delta^{13}C$ signature of all three stalagmites is higher relative to modern and Holocene speleothems (-10 to -7‰),

reflecting a signal that is buffered by the isotopic composition of the host rock (Spötl et al., 2004). High and commonly positive $\delta^{13}C$ values indicate no significant input of organic C into the system, thereby indicating an

absence of soil and vegetation above the cave throughout the duration of speleothem deposition (Spötl et al., 2008). Cooler surface temperatures and the absence of soil argue for a significantly larger Hintertux Glacier during MIS 7 relative to today, likely covering a large portion of the cave. The growth of speleothems during cold climate periods of MIS 7 is likely due to the warm-based nature of the glacier, which provides a supply of melt water while preventing the cave from freezing.

## 5 Discussion

The new Spannagel $\delta^{18}$O record spans the period of Northern Hemisphere warming associated with the later portion of TIII, the five substages of MIS 7, TIIIa, and the MIS 7/6 glacial inception (Fig. 3). Due to our unprecedented age control, we can determine the precise timing of regional changes associated with each MIS 7 sub-stages, as well as the onset of warming in the Austrian Alps associated with TIII and TIIIa (Fig. 4). While terrestrial records cannot directly date changes to the ocean-cryosphere system during a glacial termination, the climatic excursions in high-sensitivity regions, such as the Austrian Alps, provides key temporal constraints on the climate events leading up to and during these transitions.

Following the start of speleothem growth at 247.3 ±0.2 ka, an abrupt shift towards higher $\delta^{18}$O values at 242.5 ±0.3 ka marks the onset of regional warming associated with the TIII deglaciation. The ensuing interglacial period (MIS 7e) is characterized by high $\delta^{18}$O values (greater than -10‰) and spanned 5 ky from 241.8 to 236.0 (±0.3) ka. Depleted $\delta^{18}$O values (less than -12‰) between 234.3 and 216.9 (±0.3) ka coincide with MIS 7d. Maximum regional cooling occurred between 231.3–228.6 (±0.2) ka. An abrupt shift towards higher $\delta^{18}$O values at 216.8 ±0.3 ka marks the onset of regional warming associated with TIIIa. Two periods of high $\delta^{18}$O values between 215.7–212.9 (±0.4) ka (>-10‰) and 201.8–197.1 (±0.5) ka (-8.7‰) coincide with interglacial periods MIS 7c and 7a, respectively. A final shift towards lower $\delta^{18}$O values from 197.1 to 191.4 (±0.3) ka coincides with the MIS 7/6 transition, the latter portion of which is replicated by stalagmites SPA 146 and 183.

On millennial timescales, remarkable similarities are observed between Spannagel $\delta^{18}$O and paleorecords that are sensitive to the North Atlantic climate (Fig, 4). These similarities highlight the rapid climatic link between the European Alps and North Atlantic realm, as observed in later interglacial periods (e.g. Holzkämper et al., 2004; Mangini et al., 2007; Wilcox et al., 2020) and glacial periods (e.g. Moseley et al., 2014; Mayr et al., 2019). Most

striking is the millennial-scale covariance of Spannagel $\delta^{18}O$ and Chinese Monsoon $\delta^{18}O$, which is explained through the following teleconnections: temperature anomalies in the North Atlantic region influence the intensity of heat transport by northern Hadley Cell circulation, which triggers a latitudinal shift in its ascending branch, known as the Intertropical Convergence Zone (ITCZ). Latitudinal shifts in the ITCZ, in turn, influence the Chinese

Monsoon strength (see Cheng et al., 2016 for details). Thus, both Chinese and Spannagel speleothems respond to common climate forcings on millennial timescales. Identifying the mechanisms behind North Atlantic-forced excursions in Spannagel $\delta^{18}O_c$ is challenging due to the complex array of processes that influence Spannagel $\delta^{18}O$, but is likely linked to synchronous temperature changes and/or latitudinal shifts in the westerlies. In alignment with previous work, we interpret Spannagel $\delta^{18}O_c$ as a faithful recorder of millennial-scale changes in the North

Atlantic realm during MIS 7.

For the remainder of this discussion, we will examine the variations in Spannagel $\delta^{18}O$ associated with MIS 7 in order to provide new temporal constraints on climate changes in the Austrian Alps, as well as new insights into millennial-scale changes in the North Atlantic leading up to TIII and TIIIa.


**5.1 Termination III**

The last four main glacial terminations can be separated into two categories: those that were interrupted by Northern Hemisphere stadial events (TI and TIII) and those that were uninterrupted or minimally interrupted (TII and TIV) (Cheng et al., 2009). During TI, multiple meltwater pulses sourced from the decaying Northern Hemisphere ice

sheets resulted in a stratification of surface waters and expansion of winter sea ice in the North Atlantic realm (Denton et al., 2010). The expansion of sea ice amplified seasonality, such that Europe experienced cold and arid winter conditions (e.g. Renssen and Isarin, 2001).

Peak concentrations of ice-rafted detritus (IRD) in North Atlantic sediments indicate that TIII, similar to TI, was

punctuated by two discharge events, S8.2 and S8.1 (Fig. 4; Channell et al., 2012). Well-dated Spanish speleothems constrain the timeline of these events, starting with S8.2, at 249–247.4 ka (Pérez-Mejías et al., 2017). The onset of speleothem deposition in Spannagel Cave coincides with the end of the S8.2 event (Fig. 4). We interpret the lack of deposition during and prior to the S8.2 event as possible evidence for freezing conditions in Spannagel Cave during stadial conditions. Following the S8.2 event, a resumption of warmer North Atlantic conditions contributed

to increased humidity in Spain (248 ±2 ka; Pérez-Mejías et al., 2017), an abrupt strengthening of the Chinese Monsoon (247.6 ±0.9 ka; Cheng et al., 2009), and above-freezing temperatures in the central Alps prompting speleothem growth (247.3 ±0.2 ka).

A second discharge event (S8.1) occurred at 244.7–241 ka and coincided with a depletion in Spanish speleothem
$\delta^{18}O_c$ (Pérez-Mejías et al., 2017), which is interpreted as the sudden arrival of meltwater (depleted $\delta^{18}O$) into North Atlantic intermedial latitudes similar to processes observed during TI. The depleted $\delta^{18}O$ signal was likely transported downstream to the Austrian Alps. The short negative excursion in Spannagel $\delta^{18}O_c$ starting at 242.6 ±0.3 ka may represent the muted signature of this event. Following the event, Spannagel $\delta^{18}O$ exhibits an abrupt increase between 242.5 to 241.9 (±0.3) ka. This major shift in Spannagel $\delta^{18}O$ coincides with remarkable precision
with well-dated records that are sensitive to North Atlantic climate changes, including vegetation productivity in the Iberian Peninsula (241.6–240.7 ±1.6 ka; Pérez-Mejías et al., 2017) and Chinese Monsoon intensity (242.8–241.01 ±0.9 ka; Cheng et al., 2009, 2016). The Spannagel timing of TIII additionally coincides (within uncertainties) with an abrupt warming of sea surface temperatures (SST) in the North Atlantic (Martrat et al., 2007) and Mediterranean (Fig. 5; Martrat et al., 2004). Combined, these globally distributed records point to a rapid
warming in the North Atlantic realm at this time. We interpret the ~3‰ increase in Spannagel $\delta^{18}O_c$ as an abrupt rise in local winter temperatures. A possible increase in the advection of isotopically enriched Mediterranean moisture to the Austrian Alps may have amplified this warming signal, although the extent to which Mediterranean-sourced precipitation influences the $\delta^{18}O$ signature at our study site is still unclear. Using our high-precision chronology, we assign the onset of warming to 242.5 ±0.3 ka.

The Spannagel $\delta^{18}O$ record indicates a 7.6 ±0.3 ky lag in the onset of regional warming relative to the rise in 65°N summer insolation associated with TIII (Berger, 1978). The observed lag is similar but greater than the 5.1 ±0.9 ky lag in regional warming relative to TII, as recorded in a speleothem from Hölloch on the northern rim of the Alps (Moseley et al., 2015). A likely explanation for a longer lag-time may be the low obliquity forcings during TIII,
resulting in lower-than-average insolation during boreal summers which may have delayed warming in the Alps until near-peak insolation.

## 5.2 MIS 7e-d

Following TIII, a period of high Spannagel $\delta^{18}O$ values (-10‰) between 241.8 and 236.0 (±0.3) ka mark warmer
temperatures in the Austrian Alps associated with MIS 7e (Fig. 3). This interval coincides with increased humidity
in the Iberian Peninsula (Pérez-Mejías et al., 2017) and Italy (Columbu et al., 2019), and an expansion of forests in
Greece (Tzedakis et al., 2006). Spannagel $\delta^{18}O$ values reach a maximum at 240.5 ±0.3 ka, coinciding with peak
65°N summer insolation at 241.0 ka (Berger at al., 1978). The end of MIS 7e is characterized by a slow decline
starting at 236.0 ±0.3 ka, followed by an abrupt drop (-9.9 to -12‰) between 236.0 and 234.3 (±0.3) ka. This period
coincides with a steady decline of vegetation productivity in the Iberian Peninsula starting at 238.4 ±2 ka (Pérez-
Mejías et al., 2017) and a shift towards cooler conditions between ~237 and 239 ka in southeastern Europe
(Tzedakis et al., 2004). Sardinian stalagmites, however, indicate persistent Mediterranean advection (as suggested
by humid conditions) up to 230.1 ±1.6 ka (Fig. 5; Columbu et al., 2019). A decoupling of Spannagel and Sardinian
$\delta^{18}O$ values at this time further underscores that temperature, not moisture source, was the primary driver of
Spannagel $\delta^{18}O$ depletion at this time.

Depleted Spannagel $\delta^{18}O$ values (<-12‰) from 234.3 to 216.9 (±0.3) ka coincide with MIS 7d (Fig. 3). Depleted
Spannagel $\delta^{18}O$ values during this time can be attributed to cooler local temperatures, although uninterrupted
stalagmite deposition indicates that temperatures remained above freezing in this cave (Spötl and Mangini, 2007).
Maximum cooling (<-13‰) occurred between 231.3 and 228.6 (±0.3) ka. Maximum cooling coincides with the
lowest 65°N summer insolation value (387 W/m$^2$) over the last 800 ka, centered at 230.0 ka (Berger, 1978).
Maximum cooling in the Alps also coincides with an abrupt weakening of the Chinese Monsoon within ~1 ka
uncertainties (Cheng et al., 2009), suggesting Northern Hemisphere-wide cooling. During this time, sea level fell
between -18.5 m and -21 m relative to modern levels throughout (Dutton et al., 2009), atmospheric $pCO_2$ dipped
below 203 ppmv between 229.6 and 220.9 (±4) ka (Bazin et al., 2013), and North Atlantic SSTs fell to near glacial
levels (Martrat et al., 2007). In Europe, $\delta^{13}C$ records from the Iberian Peninsula (Pérez-Mejías et al., 2017) and
pollen records from Albania (Francke et al., 2016) and Greece (Tzedakis et al., 2006) indicate glacial-like
conditions during MIS 7d. In the Mediterranean realm, increased aridity recorded by Sardinian speleothems
between 225–221 (±5) ka (Fig. 5; Columbu et al., 2019) and Israeli speleothems at 223 ±4 ka (Bar-Matthews et al.,
2003) overlap within age uncertainties of Spannagel-determined MIS 7d. Overall, the Spannagel record provides
new age constraints for the timing and duration of maximum stadial conditions in Europe associated with MIS 7d.

Following the 2.7 ky time period of maximum cooling, Spannagel $\delta^{18}O$ values plateau at low (<-12‰) values until 216.8 ±0.3 ka. Low Mediterranean Sea levels (Dutton et al., 2009) and melt water pulses in the Black Sea (Badertscher et al., 2011) between ~230 to 217 ka support the of the presence of Eurasian ice sheets, which likely contributed to cooler temperatures and/or southerly shifted westerlies over Europe.

## 5.3 Termination IIIa

TIIIa is often referred to an "extra" termination that resulted from the collapse of MIS 7d ice sheets in response to anomalously high insolation. Similar to main glacial terminations, TIIIa is characterized by a rapid rise in North Atlantic SSTs (e.g. Martrat et al., 2007), atmospheric $pCO_2$ (Bazin et al., 2013), global benthic marine $\delta^{18}O$ (Lisiecki and Raymo, 2005), and an abrupt rise in sea level (e.g. Dutton et al., 2009). However, due to large age uncertainties, the exact timing of TIIIa in marine and ice records remains unclear. To resolve this issue, the timing of TIIIa has been previously determined by precisely dated Chinese stalagmites, which reveal millennial-scale weak monsoon intervals that correspond to meltwater discharge events in the North Atlantic (Cheng et al., 2009, 2016). The exact weak monsoon interval corresponding to TIIIa, however, remains a topic of debate. Cheng et al. (2009) first interpreted TIIIa as the weak monsoon interval occurring at 228 ±0.8 ka. Cheng et al. (2016) later revised this interpretation by associating TIIIa with the weak monsoon interval at 217.1 ±0.9 ka.

An abrupt rise in Spannagel $\delta^{18}O_c$ between 216.8 and 216.1 (±0.5) ka supports the timing of TIIIa defined in Cheng et al. (2016). Similar to TIII, a brief negative excursion in Spannagel $\delta^{18}O_c$ prior to the abrupt rise may correspond to the peak in North Atlantic IRD at 216 ka (Fig. 4; Channell et al., 2012) that, when aligned to our chronology, suggests that a major meltwater pulse associated with TIIIa occurred no later than 216.9 ±0.5 ka. This timing agrees within uncertainties with the Chinese monsoon weak interval at 217.1 ±0.9 ka. Our record reveals that the onset of Spannagel $\delta^{18}O_c$ increase occurred 1.2 ky after peak 65°N summer insolation. A lack of change in stable isotopes from Mediterranean-dominated records during this time (e.g. Columbu et al., 2019) further suggests the abrupt increase of Spannagel $\delta^{18}O_c$ was primarily driven by warming temperatures in the Austrian Alps (Fig. 5).

## 5.4 MIS 7c-a

High $\delta^{18}O$ values associated with MIS 7c occur between 215.7 and 212.9 ($\pm$0.4) ka, with slightly lower values ($\sim$ -10.6‰) extending until 212.0 $\pm$0.4 ka (Fig. 3). $\delta^{18}O$ drops abruptly between 212.0 and 211.7 ($\pm$0.4) ka and remains low between 211.7 and 204.1 ($\pm$0.4) ka. This period of depleted $\delta^{18}O$ values coincides within uncertainties with low 65°N summer insolation (207 ka) associated with MIS 7b. Spannagel $\delta^{18}O$ rises between 204.1 and 201.5 ($\pm$0.4) ka and remains high for the remainder of MIS 7a until 197.1 ($\pm$0.3) ka.


Maximum $\delta^{18}O_c$ values during MIS 7a-c indicate warmer winter temperatures in the Alps relative to MIS 7e, with MIS 7a the warmest sub-stage. This MIS 7 sub-stage comparison is in agreement with planktonic foraminiferal assemblages on an Iberian Margin sediment core which suggest higher winter temperatures (+1.5°C) during MIS 7a relative to MIS 7e (Desprat et al., 2006). European pollen data indicates that MIS 7c-a had the longest duration,

the most diverse and complete forest succession, and the warmest temperatures relative to MIS 7e (Penaud et al., 2008; Tzedakis et al., 2004). Enriched Spannagel $\delta^{18}O_c$ values during MIS 7a-c may have been further amplified by an increase in northerly advection from the Mediterranean, as suggested by humid conditions in Sardinia (Columbu et al., 2019) and eastern Mediterranean sapropel deposits (e.g. Ziegler et al., 2010), although the extent to which Mediterranean-sourced precipitation influences the $\delta^{18}O$ signature at our study site is still unclear.


Spötl et al. (2008) proposed that four fluorescent inclusions (i.e. dust layers) identified in SPA 121 reflect a partial retreat (or repeated partial retreats) of the Hintertux glacier. Our updated chronology places the timing of the dust layers at 214.3 $\pm$0.4 ka, which coincides with maximum obliquity forcing (Fig. 3; Berger, 1978). This finding supports previous work which argues that changes in obliquity act as a major control on alpine glacier mass balance

by influencing the latitudinal distribution of solar radiation (e.g. Huybers et al., 2006). High obliquity and insolation forcing likely drove increased ablation of the Hintertux glacier and warm regional temperatures (as suggested by maximum MIS 7c $\delta^{18}O_c$ values) at 214.3 $\pm$0.4 ka.

## 5.5 MIS 7-6 transition

A drop in Spannagel $\delta^{18}O$ starting at 197.1 $\pm$0.2 ka marks the end of MIS 7a in the Alps (Fig. 3). We define the Spannagel MIS 7/6 transition to between 197.1–191.4 ($\pm$0.3) ka, which coincides with a drop of the Mediterranean

Sea level, as shown by Bard et al. (2002). Two newly collected stalagmites from this cave (SPA146 & 183) that grew between 191.8 and 182.3 (±0.6) ka provide additional records of the late MIS 7a/6e transition and early MIS 6e. All three stalagmites show a gradual decreasing trend in $\delta^{18}O_c$ until approximately 187 ka. A lack of calcite

deposition after 182.3 ±0.2 ka suggests unfavorable conditions in Spannagel Cave, possibly related to due to partly cold-base conditions of the glacier above the cave.

**6 Conclusions**

The response of climatically sensitive regions to glacial-interglacial cycles and their abrupt transitions provides

key insight into the timing of global climate change. In this study, we present the first ever paleo-record of MIS 7 with relative age uncertainties <2‰. Using this chronology, we can determine the precise timing and duration of climate variations in the Austrian Alps in response to the MIS 7 sub-stages and associated glacial terminations. Following the start of speleothem growth, an abrupt increase of $\delta^{18}O$ values at 242.5 ±0.3 ka marks the onset of regional warming associated with TIII. The ensuing interglacial period (MIS 7e) is characterized by enriched $\delta^{18}O$

values (> -10‰) from 241.8 to 236.0 (±0.3) ka. Depleted $\delta^{18}O$ values (< -12‰) between 234.3 and 216.9 (±0.3) ka coincide with the Northern Hemisphere cool period. Similar to TIII, a brief negative excursion in Spannagel $\delta^{18}O_c$ prior to the abrupt rise associated with TIIIa suggests that a major meltwater pulse occurred in the North Atlantic around 216.9 ka. An abrupt shift towards higher $\delta^{18}O$ values at 216.8 ±0.3 ka marks the onset of regional warming associated with TIIIa. Two periods of high $\delta^{18}O$ values between 215.7–212.9 (±0.4) ka and 201.8–197.1 (±0.5) ka

coincide with interglacial periods MIS 7c and 7a, respectively. A final shift towards lower $\delta^{18}O$ values from 197.1 to 191.4 (±0.3) ka and coincides with the MIS 7/6 transition. In total, this multi-stalagmite record provides important chronological constraints on climate shifts in the Austrian Alps associated with MIS 7, while providing new insight into the timing of millennial-scale changes in the North Atlantic realm.

**Author contribution**

C.S. collected the samples. K.W. and X.L. conducted measurements and analyzed results. H.C., R.L.E., and. C.S. provided scientific guidance, laboratory facilities, and funding. All authors contributed to the final manuscript. Special thanks to M. Wimmer and M. Pythoud for their assistance in the laboratory.

## Acknowledgments

This research was partly funded by grants from the Austrian Science Fund (FWF) awarded to C.S., NSFC 41888101
to H.C., and NSF1702816 to RLE.

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

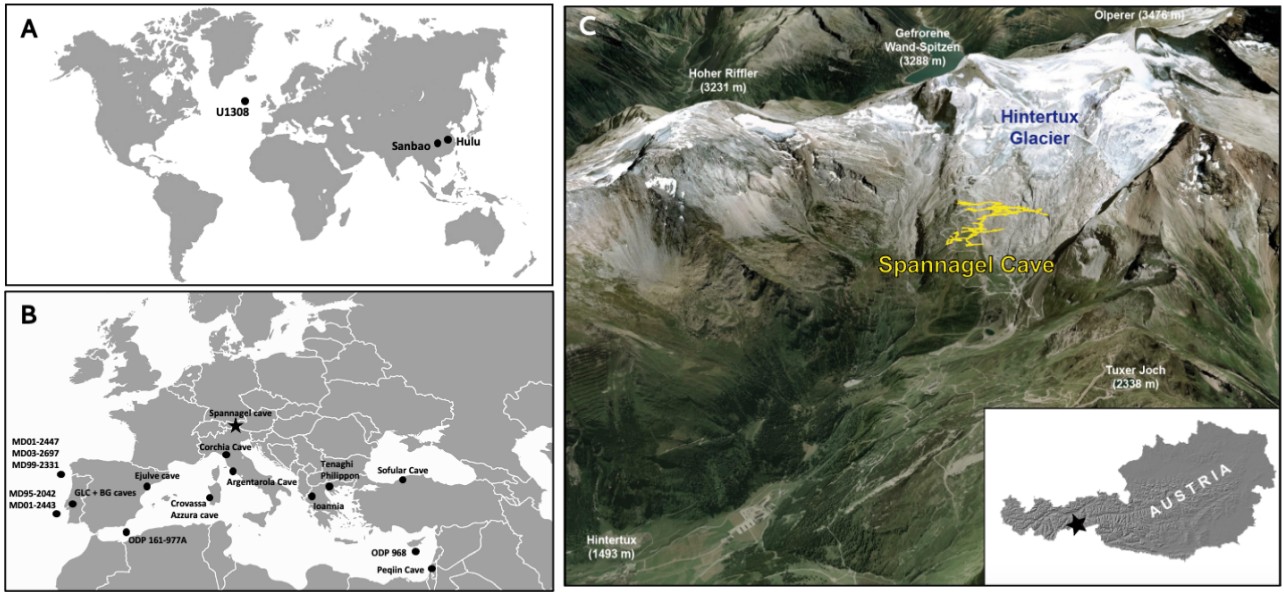

**Figure 1: (A-B) Location of paleorecords described in this paper. (C) Google Earth image of the Tux Valley showing the location of Spannagel cave (yellow) adjacent to the glaciated main ridge of the Alps (oblique view towards SSE) © Google Earth.**


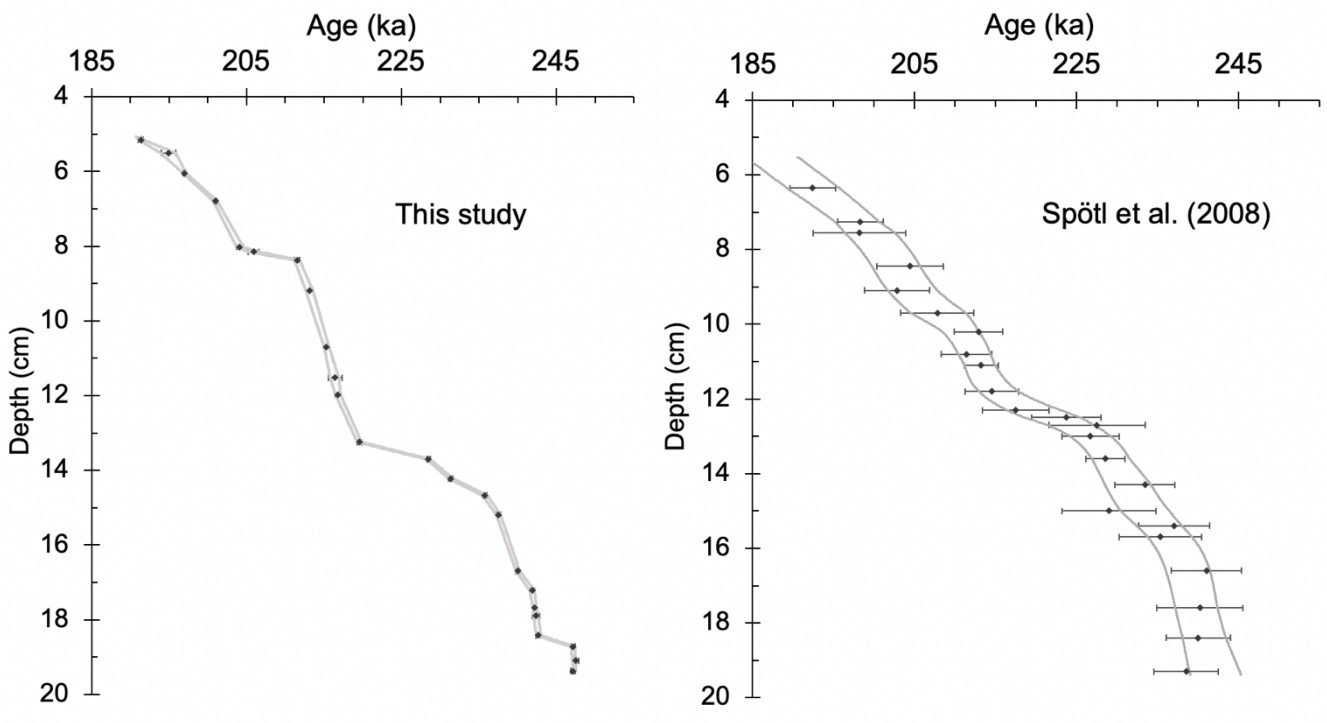

**Figure 2: Depth versus age along the growth axis of stalagmite SPA121 with associated 2 sigma age uncertainties from this study (left) and Spötl et al. (2008) (right). Grey lines show upper and lower 2σ uncertainties of each age model.**


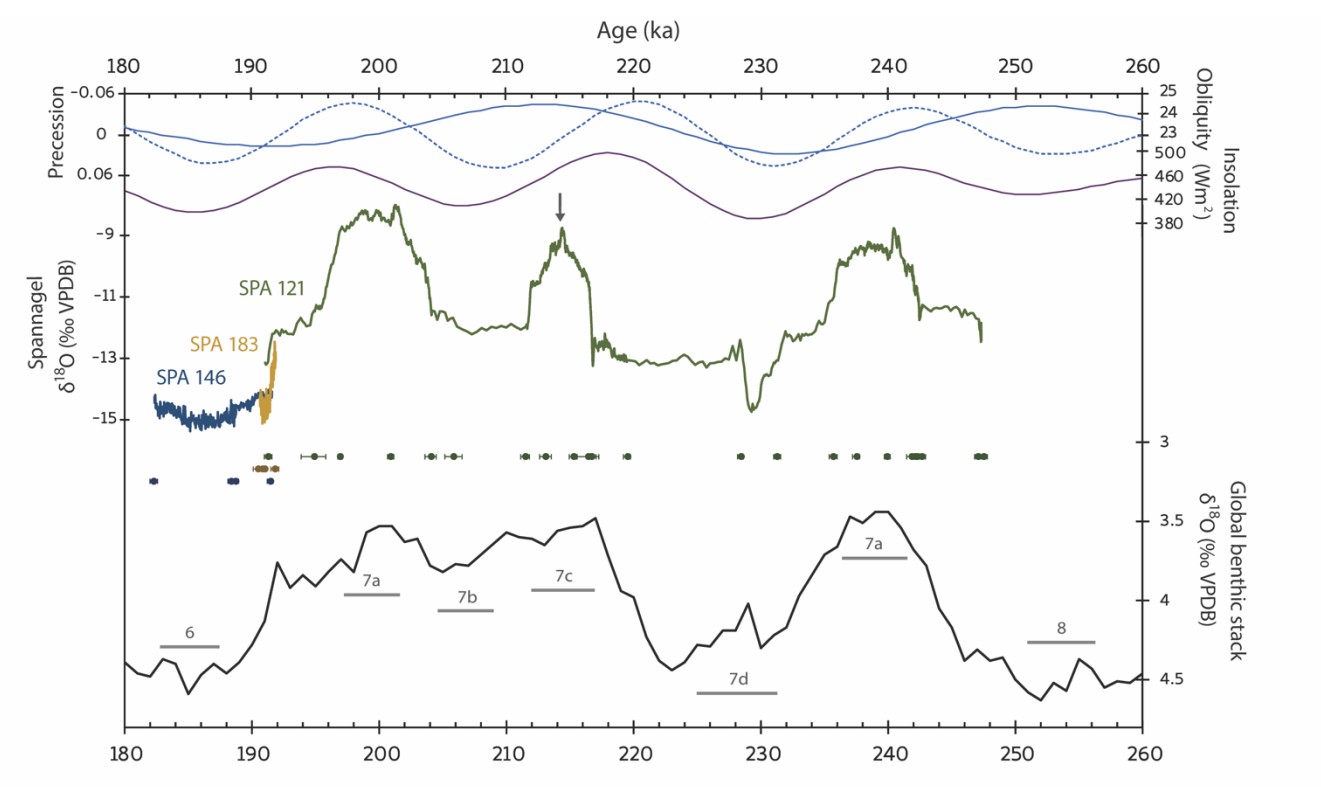

**Figure 3: Obliquity (blue), precession (dashed blue), and 65°N July insolation (dark purple) from Berger (1978). Spannagel δ¹⁸O from stalagmites SPA121 (green), SPA183 (yellow), and SPA146 (dark blue) (this study). Global stacked benthic δ¹⁸O (black; Lisiecki and Raymo, 2005) plotted with MIS substage labels (grey) for reference. Arrow represents the age of fluorescent inclusions which suggest a partial retreat of the Hintertux glacier (214.3 ±0.4 ka).**

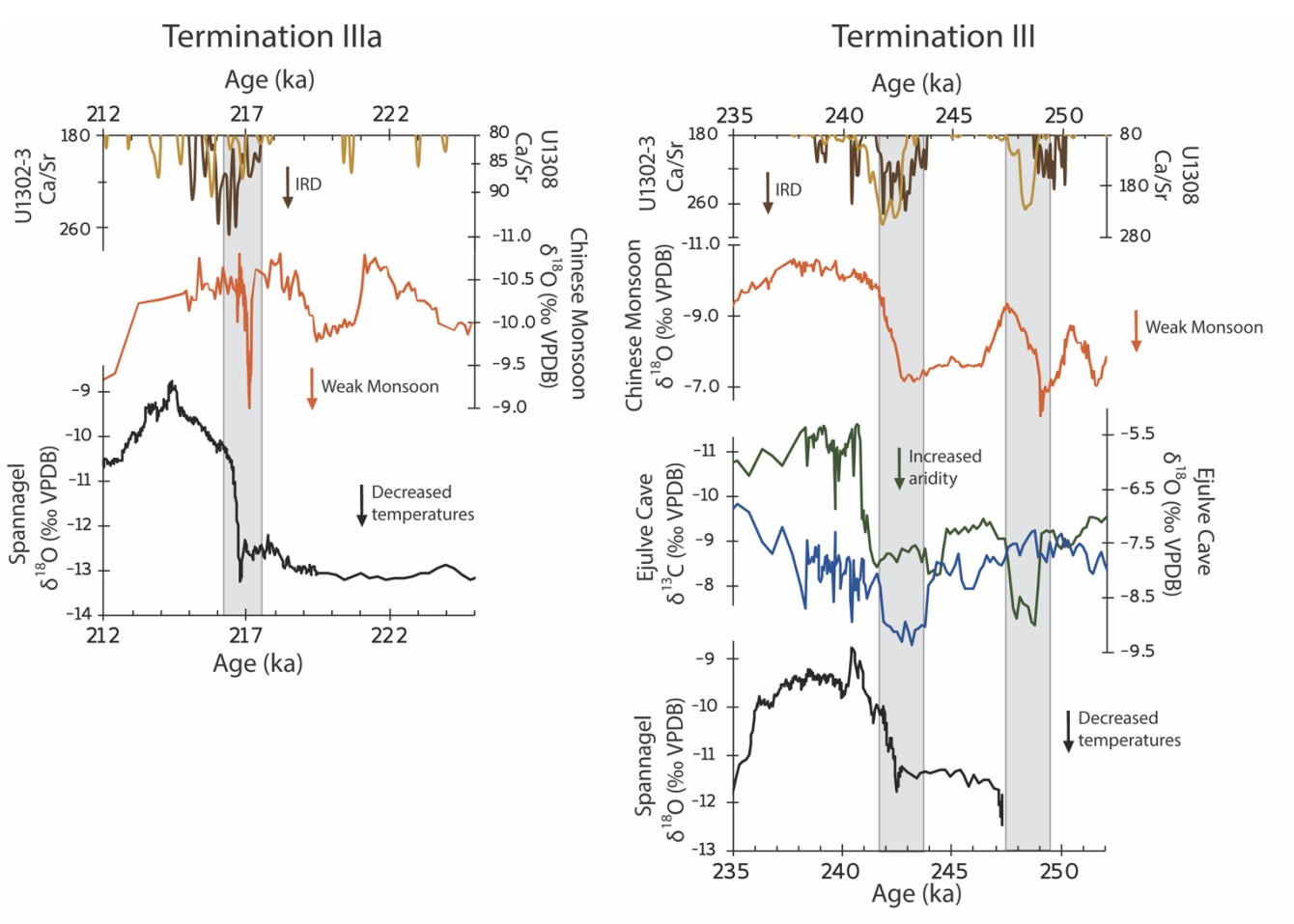

Figure 4: Millennial-scale events leading up to glacial terminations III and IIIa. Ca/Sr ratios (a proxy for IRD) from North Atlantic sediment cores U1308 (brown; Channell et al., 2012) and U1302 and U1303 (yellow, Hodell et al. 2008), Chinese stalagmite δ18O (orange, Cheng et al., 2016), Ejulve cave δ18O (blue) and δ13C (green) from southeastern Spain (Pérez-Mejías et al., 2017) and Spannagel Cave δ18O from the Austrian Alps (black; this study). Major IRD events highlighted in grey.


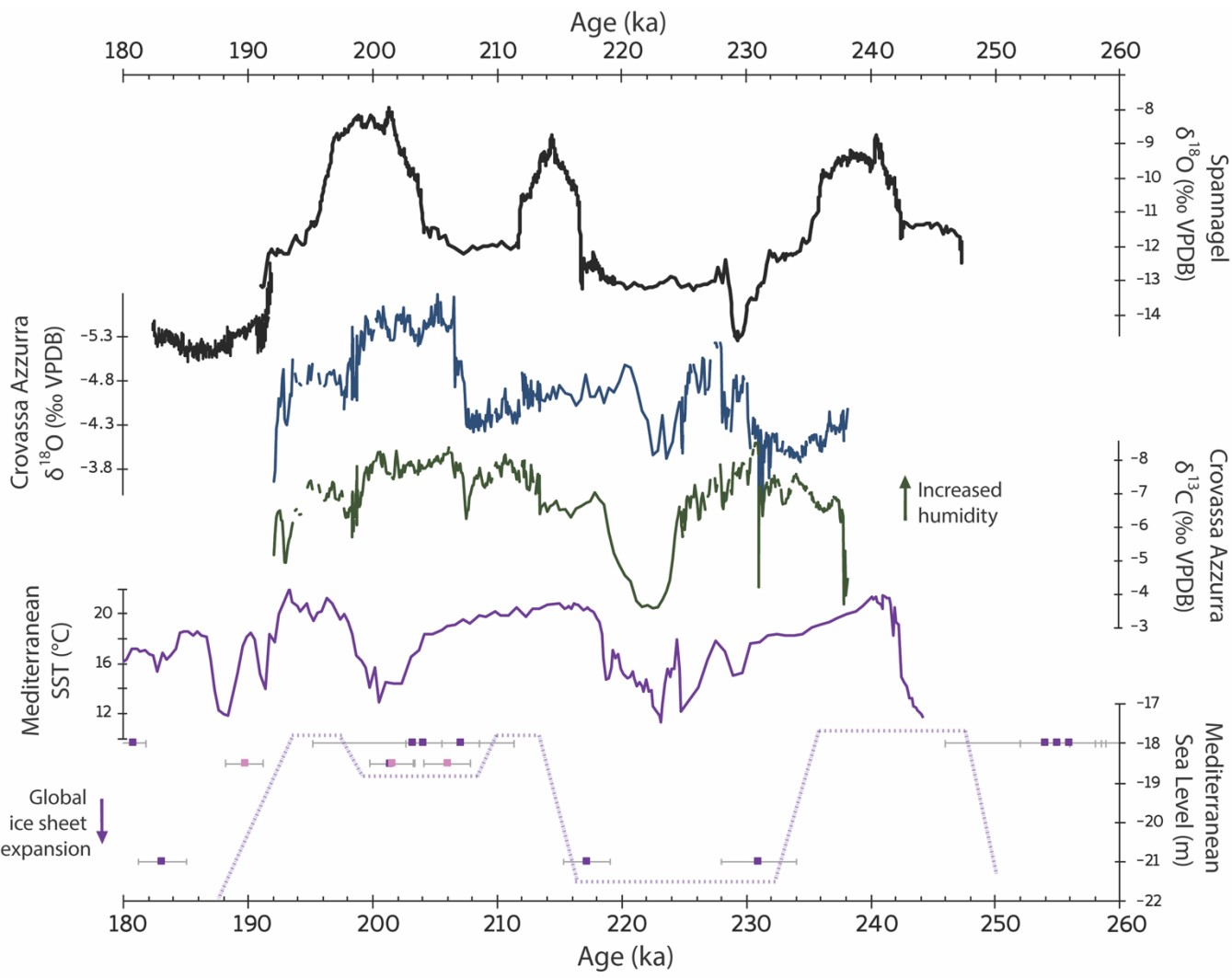

**Figure 5: MIS 7 records from the circum-Mediterranean region.** Spannagel Cave δ¹⁸O (black; this study), Crovassa Azzurra (Sardinia, Italy) δ¹⁸O (blue) and δ¹³C (green) (Columbu et al., 2019), Western Mediterranean SST (Martrat et al., 2004), and Mediterranean sea-level reconstructions (purple; Dutton et al., 2009) and (pink; Bard et al., 2002). Purple dashed lines are for visual aid only; absolute sea levels are unknown. Spannagel δ¹⁸O events that are decoupled from Mediterranean temperatures and humidity records represent evidence for temperature-driven δ¹⁸O excursions in the Austrian Alps, as opposed to changes in the proportion of Mediterranean-sourced moisture arriving at our study site.