# Peer review of "Precise timing of MIS 7 sub-stages from the Austrian Alps"

_Climate of the Past, 2020_

## Referee Comment (RC1) · Anonymous Referee #1 · 3 Dec 2020

General comment

This manuscript offers data of great interest to our community since it provides - with excellent precision - the timing of TIII and different substages of MIS7 interval. This time interval is not well represented in terrestrial records all over the world and from marine records obtaining an independent chronology is not easy for that period. Then, the novelty and interest of this study are assured. The data come from three different speleothems from the same cave, Spannagel Cave in Central Alps; two of the stalag-mites are new ones (SPA146 & 183) and cover MIS7/6 transition and the other one (SPA-21) has been certainly improved in its chronology (previous age model in Spötl et al., 2008) and covers MIS7 in its totallity. As a general comment, I consider this manuscript suitable to be published in CP with some moderate changes. I suggest,

focusing on the SPA-21 stalagmite and MIS7 chronology and excluding the other two speleothems (if not excluded, at least modulate the sentences regarding replication-see below).

Specific comments

1.- Replication of the three stalagmites. I am not very convinced of the data presented from the two new stalagmites... and I think the authors should consider the benefits of including them here. Amplifying Fig. 3, I am not completely sure if both new stalagmites show the rapid change in d18O at 192-191 ka, as SPA-21 does (I don't think so, but it is difficult to evaluate in the figure). On the other hand, d13C values are certainly not replicated. Therefore, taking into account that the SPA-21 signal is not replicated by SPA146 and SPA183 records, since those two stalagmites start once SPA-21 stops growing (Fig. 3), I would suggest the authors to consider the advantage of including them in the study. Additionally, SPA146 and SPA183 present some issues close to the base of their MIS7 section (see Fig. S2) and I am afraid that those problems related to diagenetic alteration may still influence above that section. My comment is also based on the high correlation among d18O and d13C in the provided Hendy tests (Fig. S3) that is not present in SPA-21 (Spotl et al., 2008) thus suggesting kinetic effects. In addition, the age model of these two stalagmites from 197.1 to 191.4 ($\pm$0.3) ka is not present in the manuscript... that is certainly important to consider those stalagmites here.

In any case, if the authors still maintain SPA146 and SPA83 data in the paper, they have to provide their age models and avoid statements such as that one from line 165 ("....final shift towards lower $\delta$18O values from 197.1 to 191.4 ($\pm$0.3) ka and coincides with the MIS 7/6 transition, the latter portion of which is replicated by stalagmites SPA 146 and 183") since I don't see any portion replicated. Or line 140 ("Hendy tests on the new stalagmites SPA146 (averaged R2=0.88) and SPA183 (averaged R2=0.84) indicate possible kinetic influences (see Fig. S3), although the replication in the general trend and absolute values of all three stalagmites argue against significant kinetic ef-
fects") since there is not a replication in the general trend or absolute values in the three stalagmites.

2.- Source of precipitation. The authors indicate two main sources of precipitation in this region, which can be differentiated by the d18O isotopic values. I agree with this statement, but I consider that Atlantic sourced precipitation may not be much more negative than the Mediterranean one, depending on the moisture uptake along the longer pathway. Rainout effect is sometimes compensated by the more positive recycled moisture that is being incorporated in the way from the source to the Central Alps. It is then important to take into account the moisture recharge through the long pathway as, sometimes, the result is an enrichment derived by the effect of enriched inland moisture compared to ocean moisture. See, for example, Chakraborty et al., (2016) and Krklec and Domínguez-Villar (2014). More references on this topic or an additional study of moisture sources in the Central Alps may be of interest to clearly ascribe the Atlantic source with more negative d18O values.

3. Similarity with d18O monsoon records. The authors indicate several times in the discussion the high similarity with Asian monsoon records (lines 175, lines 235, etc); I think these statements should be modulated as I observe many differences in timing and pattern in Fig. 4. Both the similarities and the differences must be clearly described. For example, the time of TIIIa is completely different, also the pattern. The time of 7d as defined in Spanagel (234-216 ka) does not coincide at all with Chinese monsoon timing. Please indicate and explain potential mechanisms for those differences.

Minor comments

- Line 26. I miss one or two references here to support this statement.

- Line 140. Replication just happens during very short periods of time, if any, and the values and trends are not so well reproduced. I would not use those criteria for discarding kinetic effects.

- Line 147. This just applies for SPA21, the other two stalagmites display more negative values. Please, explain why.

- Figure 4. I would suggest adding to this figure the duration of MIS7 substages (lines or shaded squares) to really see when they start and finish, not only the "peak" indicated by the name in Fig. 4D.

- Figure S3. I think these data correspond to two different lamina in every stalagmite. Please indicate it in the graph or caption.

References

Chakraborty, S., Sinha, N., Chattopadhyay, R., Sengupta, S., Mohan, P. M. and Datye, A.: Atmospheric controls on the precipitation isotopes over the Andaman Islands, Bay of Bengal, Scientific Reports, 6(1), doi:10.1038/srep19555, 2016.

Krklec, K. and Domínguez-Villar, D.: Quantification of the impact of moisture source regions on the oxygen isotope composition of precipitation over Eagle Cave, central Spain, Geochimica et Cosmochimica Acta, doi:10.1016/j.gca.2014.03.011, 2014.

Spötl, C., Scholz, D. and Mangini, A.: A terrestrial U/Th-dated stable isotope record of the Penultimate Interglacial, Earth and Planetary Science Letters, 276(3–4), 283–292, doi:10.1016/j.epsl.2008.09.029, 2008.

---

## Referee Comment (RC2) · Anonymous Referee #2 · 16 Dec 2020

This article proposes new radiometric ages for glacial-interglacial and stadial-interstadial transitions across marine isotope stage (MIS) 7. The durations of these transitions are also presented. To date, few records offer precise ages for these events, so this ms is potentially of much interest. In summary, the main time series presented (SPA121) comprises new age data applied to a previously published isotope record. The new ages significantly improve the precision of the SPA121 time series, which is a useful advance. The series from the two other stalagmites provide additional data that support the authors' claim that the composite record reaches into MIS6 and therefore covers the MIS7a-MIS6 transition.

The main issue I have with this ms is the authors' claim that the sharp speleothem oxygen isotope (18O) changes (both increases and decreases) can be used to date

the various MIS 7 transitions. To me, this is not convincingly demonstrated because I feel the use of the terms 'MIS 7e-d transition', 'termination 3', etc. is not done with full recognition of what these important terms actually mean. I will use Termination 3 as an example. Put simply, a Pleistocene termination is a global event representing the entire period over which a large percentage of Earth's cryosphere decays (mostly Northern Hemisphere ice sheets). Termination 1 started at around 20 ka (Denton et al 2010 Science) and spanned well over 10,000 years. Previous radiometrically based records of T3 (Cheng et al. 2009, 2016; Pérez-Mejías et al. 2017) imply shorter durations but no where near <1 kyr. Here, Wendt et al. state that T3 occurred over just ca. 700 years (table 1). Besides the potential physical impossibility for melting so much ice so quickly, this short time frame also means very large adjustments to existing age scales for marine time series (incl. LR04) covering this period (I don't just mean wholesale shifts of the ocean record to older or younger U-Th based ages of events – this is to be expected because of limited age control in these records – but the extreme squeezing and stretching of sedimentation rates to unrealistic levels). A 700-yr termination would also drastically change the time scale of Antarctic ice cores, which although having inherent uncertainties of its own also has it limits in terms of how much accumulation rate and ice-flow modelling change can be tolerated. Termination 3a is a similar story: it is also argued to have been completed in just ca. 700 years.

In my opinion, the authors misuse the terms 'transition' and 'termination'. The point I want to make is that speleothems do not preserve terminations or other MIS transitions per se. Ocean sediments do. Speleothems (and other archives) preserve the local or regional expression of climate changes associated with these transitions. Therefore, in assigning ages of MIS transitions using a speleothem chronology one must first re-solve how the climate signal in the speleothem actually records such transitions and how these are linked to relevant ocean record(s). The authors do refer to ocean records in the ms (LR04, MD01-2444: Figure 4) but do not determine exactly how the Alpine speleothem 18O profile links to these records, apart from references to alpine warming coeval with SST increases (and the converse). To cite Termination 3 again. . . Regarding the interval of stable d18O values between 247 and 242 ka: to which part of the ocean record does this correspond? Is it the 'late MIS8 glacial' before the termination actually starts (it would seem so, based on the authors' claims of a short termination that starts after this isotope plateau), or is it really part of the period of ice-sheet melting associated with the termination, as implied in Cheng et al. 2016 and 2009, and Pérez-Mejías et al. 2017? If the latter, which in my opinion (based on all the evidence) is more realistic, the quoted ages and durations of T3 and potentially other transitions listed in table 1 have little meaning. The case for a link between the cave and ocean records through the whole time interval must be better developed. This needs to bear in mind that the age and duration of a given climate transition is global and based essentially on changes in global ice volume, and therefore is best resolved in ocean records. A speleothem (or lake record, etc.) will respond to this event according to local and regional climate dynamics. From what I can determine, it seems that the speleothem did not even capture all of T3, if you take into consideration previously published speleothem records (Cheng et al. 2009 and 2016 and Pérez-Mejías et al. 2017). It obviously captures all of 7e, the 7e/d transition and the 7d/c transition (T3a), etc. but exactly how do the boundaries of these transitions in the ocean record tie to the speleothem 18O?

In the context of the above, I would like the authors to carefully consider exactly what the abrupt speleothem 18O changes mean at this high altitude cave? For instance, are the abrupt increases examples of Bølling-Allerød-like or YD-Holocene-like events? Hard to say – age uncertainties, although small in percentage terms, are still too large to test whether the true duration of these events are comparable. But this is tantalising and really important because it implies that T1 was not alone with its two rapid NH temperature jumps, and that T3 likely had at least one comparable rapid warming (at least in this part of the N Hemisphere) well after it started. We know from T1 that the BA transition occurred ~5 kyr into the termination.

There is an alternative explanation the authors should consider too: is the speleothem

18O acting like an 'on-off' switch, i.e. does it represent binary swings between (i) periods when the glacier is present above the cave (when basal meltwaters with low 18O values derived from strongly 18O-depleted glacial or stadial snowfall occurring 1000-1200 m higher than the cave itself, near the Hintertux glacier summit ∼3500 m a.s.l.) and (ii) periods when the glacier retreats during interglacials and interstadials and exposes the cave recharge area to direct infiltration (at ∼2300 m) of isotopically enriched rainfall and in situ snowfall? This could explain the almost square-wave form and amplitude of the speleothem isotopic series (otherwise for the MIS7a-MIS6 transition, for example, we must consider 20 deg C or more of temperature depression plus a little extra for possible changes in moisture source, given the >6 per mil decrease in speleothem 18O). This raises the question of whether the sharp increases and decreases in d18O are really a local effect of ice retreat, whose phasing with respect to regional warming and cooling (e.g. the rises and falls in SST in MD01-2444) is not as closely coupled as the authors think.

This all sounds rather negative, but I encourage the authors to re-consider the meaning of MIS transitions, as global events with local and regional expressions. This is exceptional radiometric dating – I hope the authors can apply these precise results it in a more meaningful way.

---

## Author Comment (AC1) · 5 Mar 2021

We greatly appreciate the excellent insight provided by reviewer #1. Below is a list of individual comments and questions followed by our responses:

1. I am not very convinced of the data presented from the two new stalagmites, and I think the authors should consider the benefits of including them here.

We agree that our original manuscript lacked sufficient evidence to ensure that the two new stalagmites (SPA 146 and 183) were deposited close to isotopic equilibrium with dripwaters. To address this issue, we have re-sampled 8 Hendy tests from SPA146 and 5 from SPA183 (see new supplementary figures, attached). The results present a more comprehensive picture of the two stalagmites. The lack of statistically significant

covariation suggests that the isotope records of SPA 146 and SPA 183, like SPA 121, are not significantly affected by kinetic processes.

It's important to note that the stable isotope values from each stalagmite do not replicate each other perfectly – a feature commonly seen in speleothem studies elsewhere. This may be due to several factors (see responses below for a discussion on this topic). However, we argue that the isotopic trends generally agree and can be interpreted in relative terms as a proxy for past regional climate changes. To this end, we maintain that stalagmites SPA 146 and 183 provide valuable insight into the timing of and regional cooling at the onset of MIS 6.

2.- Source of precipitation. The authors indicate two main sources of precipitation in this region, which can be differentiated by the d18O isotopic values. I agree with this statement, but I consider that Atlantic sourced precipitation may not be much more negative than the Mediterranean one, depending on the moisture uptake along the longer pathway. Rainout effect is sometimes compensated by the more positive recycled moisture that is being incorporated in the way from the source to the Central Alps. It is then important to take into account the moisture recharge through the long pathway as, sometimes, the result is an enrichment derived by the effect of enriched inland moisture compared to ocean moisture. See, for example, Chakraborty et al., (2016) and Krklec and Domínguez-Villar (2014).

A discussion of precipitation d18O in the central Alps is provided on lines 57-62. On line 57 we cite a modern back-trajectory study. On line 60 we cite a study that examined the various processes that influence d18O signatures of modern precipitation in the central Alps.

A discussion of the d18O values of dripwaters in Spannagel cave is provided on lines 79-82. We cite studies that investigated the d18O signature of Spannagel dripwater and its comparison to local precipitation sourced from both Atlantic and Mediterranean regions. Overall, we believe that this combination of site-specific modern precipitation

and dripwater calibration studies provides sufficient evidence to support our interpretation of Spannagel d18O.

3. Similarity with d18O monsoon records. The authors indicate several times in the discussion the high similarity with Asian monsoon records (lines 175, lines 235, etc); I think these statements should be modulated as I observe many differences in timing and pattern in Fig. 4. Both the similarities and the differences must be clearly described. For example, the time of TIIIa is completely different, also the pattern. The time of 7d as defined in Spanagel (234-216 ka) does not coincide at all with Chinese monsoon timing. Please indicate and explain potential mechanisms for those differences.

We agree that our original text lacked a full-picture explanation behind the interpreted similarities between the Chinese monsoon and Spannagel records. For example, we highlighted the similarities between variations in the Spannagel $\delta$18O and Chinese Monsoon $\delta$18O on sub-orbital timescales (e.g. line 175), but did not explain why the two locations are decoupled on other timescales (e.g. orbital) which may cause confusion to readers. We have now expanded our discussion of Spannagel $\delta$18O vs Chinese Monsoon $\delta$18O.

Regarding the timing of MIS 7d: we agree with the reviewer that Fig. 4 and its related text may render confusing without a clear reiteration of age uncertainties associated with both records. For example, the period of maximum isotopic depletion of MIS 7d calcite in Spannagel (229.2 ka) appears early, but is actually within the $\sim$1 ka age uncertainties of the Chinese Monsoon record (228.2 ka). Nevertheless, statements such as line 235: "Remarkable similarities in the shape and timing of maximum MIS 7d conditions between Chinese Monsoon and Spannagel $\delta$18O provide clear evidence for abrupt cooling of the North Atlantic at this time" comes across too bold without an emphasis on the respective age uncertainties of each record. We have reworded these and similar statements to include a comparison of age uncertainties.

The end of MIS 7d is marked by the onset of TIIIa, which according to Cheng et al. 2016, occurs in the Chinese monsoon record at 217.1 ±0.9 ka (see line 254). The timing of TIIIa is therefore well within uncertainties of both records. To further emphasize this, we have added brief explanation of the timing of TIIIa in the Chinese monsoon in the caption of Fig. 4.

- Line 26. I miss one or two references here to support this statement.

We added the citation PAGES (2016) as a summary reference for MIS 7 substages.

- Line 140. Replication just happens during very short periods of time, if any, and the values and trends are not so well reproduced. I would not use those criteria for discarding kinetic effects.

We have re-sampled multiple Hendy tests for SPA146 and 183 in order to provide a more robust test for possible kinetic effects (see response to first comment). It is true that the absolute d18O and d13C values from each stalagmite do not replicate each other perfectly. This may be due to several factors (see response below). However, we argue that the isotopic trends generally agree between stalagmites, particularly in their trend towards depleted values at the onset of MIS 6. Returning to the reviewers' point, we no longer argue that replication is the main line of evidence against kinetic effects; instead, we point to the evidence provided by the new Hendy test analysis.

- Line 147. This just applies for SPA21, the other two stalagmites display more negative values. Please, explain why.

We argue that prior calcite precipitation (PCP) likely influenced the stable isotope values of SPA121. Two lines of evidence support this hypothesis. During cool periods, low d18O intervals recorded in SPA 121 coincide with very high d13C (e.g. during the MIS 7a-6 transition). During warm periods, second order features locally co-vary (see discussion section 4.3 in Spötl et al. 2008). These observations point to a kinetically controlled process, such as PCP. A second line of evidence is that SPA 121 grew very

slowly relative to SPA 146 and 183, indicating a very slow drip rate (and thus, higher likelihood of PCP).

The results of PCP would lead to an overall enrichment in the absolute values of d18O and d13C recorded in SPA 121. The growth rate of SPA 121 is consistently slow; thus, one could argue that PCP was active throughout our time period of study. Assuming this, PCP alone cannot account for the shifts in stable isotope values observed during the transition between MIS 7 sub-stages. We maintain that these shifts reflect changes in the regional climate system. We thank the reviewer for raising this question and have added a more detailed explanation to the text.

- Figure 4. I would suggest adding to this fi̧gure the duration of MIS7 substages (lines or shaded squares) to really see when they start and fi̧nish, not only the "peak" indicated by the name in Fig. 4D.

Done

- Figure S3. I think these data correspond to two different laminae in every stalagmite. Please indicate it in the graph or caption.

We have replaced Figure S3 (see attached figures and response to first comment)
* * *
[Figure]

**Fig. 1.** New Supplementary Figure A (to replace supplementary figure 3)

[Figure]

**Fig. 2.** New Supplementary Figure B (to replace supplementary figure 3)

---

## Author Comment (AC2) · 5 Mar 2021

We greatly appreciate the excellent insight provided by reviewer #2. Below is a list of individual comments and questions followed by our responses:

1. In my opinion, the authors misuse the terms 'transition' and 'termination'. The point I want to make is that speleothems do not preserve terminations or other MIS transitions per se. Ocean sediments do

We agree that our interpretation and use of the term "termination" was used too liberally in our original text. As the reviewer correctly pointed out, this paper is a study of d18O changes speleothems from the central European Alps. Although evidence suggests that Spannagel d18O is highly sensitive to changes in the North Atlantic realm,

speleothems (as with all terrestrial archives) cannot be used to directly date changes in the ocean-cryosphere system. To this end, we have revised the manuscript to avoid such sweeping statements. We also expanded the first paragraph of our discussion section (starting on line 156) to emphasize that Spannagel d18O records the regional expression of climate changes associated MIS 7 sub-stages and terminations.

2. The authors do refer to ocean records in the ms (LR04, MD01-2444: Figure 4) but do not determine exactly how the Alpine speleothem 18O profile links to these records, apart from references to alpine warming coeval with SST increases (and the converse). . . The case for a link between the cave and ocean records through the whole time interval must be better developed.

We agree that an expanded discussion of this topic is needed. North Atlantic sediments show clear evidence of SST warming and the deposition of ice-rafted debris associated with TIII and TIIIa (e.g. Martrat et al. 2007; Channell et al. 2012). The onset of these oceanic changes occurs within age uncertainties with an abrupt enrichment of d18O in Spannagel speleothems. The temporal agreement between marine and terrestrial records suggests that warming in the North Atlantic realm associated with TIII and TIIIa triggered warmer winter temperatures in the European Alps. This pattern is consistent with later terminations (e.g. Spötl et al., 2002 Geology). It is also clear, however, that Spannagel d18O does not remain coupled with North Atlantic SSTs throughout the entirety of MIS 7, such as during MIS 7d. This observation highlights the multiple driving forces that influence winter temperatures in the central Alps. We have added a discussion of the complex link between the North Atlantic and central Alps in the first paragraph of the discussion section.

3. Regarding the interval of stable d18O values between 247 and 242 ka: to which part of the ocean record does this correspond? Is it the 'late MIS8 glacial' before the termination actually starts (it would seem so, based on the authors' claims of a short termination that starts after this isotope plateau), or is it really part of the period of ice-sheet melting associated with the termination, as implied in Cheng et al. 2016 and

2009, and Pérez-Mejías et al. 2017? If the latter, which in my opinion (based on all the evidence) is more realistic, the quoted ages and durations of T3 and potentially other transitions listed in table 1 have little meaning.

We agree - our record cannot rule out the possibility that the depleted isotope plateau between 247 and 242 ka is associated the onset of ice sheet melting associated with TIII. It is therefore incorrect to define the timing and duration of TIII in the strict terms outlined in our first draft.

To avoid confusion, we have re-worded table 1 (see attached) and adjusted our wording throughout the text to emphasize that the well-dated shifts in Spannagel d18O may reflect the precise timing of climate changes in the North Atlantic realm during TIII—but that they cannot define the exact ocean-cryosphere changes forcing them.

4. From what I can determine, it seems that the speleothem did not even capture all of T3, if you take into consideration previously published speleothem records (Cheng et al. 2009 and 2016 and Pérez-Mejías et al. 2017). It obviously captures all of 7e, the 7e/d transition and the 7d/c transition (T3a), etc. but exactly how do the boundaries of these transitions in the ocean record tie to the speleothem 18O?

If one defines TIII as the period of maximum IRD deposition in the North Atlantic between 243-240 (e.g. Channell et al. 2012), then the period between the onset of speleothem deposition at 247.3±0.2 ka and the abrupt 3‰ increase of Spannagel d18O from 242.5 to 241.9 (±0.3) ka is well within uncertainties of the IRD deposition event. This time frame is also consistent with the abrupt changes with vegetation productivity in the Iberian Peninsula (241.6–240.7 ±1.6 ka; Pérez-Mejías et al. 2017) and Chinese Monsoon intensity (242.8–241.01 ±0.9 ka; Cheng et al. 2009, 2016) – both of which are interpreted by the authors as the periods of maximum North Atlantic warming associated with TIII. Spannagel may not cover all of the earlier events leading up to TIII, but we disagree with the statement that Spannagel does not fully capture TIII.

5. In the context of the above, I would like the authors to carefully consider exactly what

the abrupt speleothem 18O changes mean at this high altitude cave? For instance, are the abrupt increases examples of Bølling-Allerød-like or YD-Holocene-like events? Hard to say – age uncertainties, although small in percentage terms, are still too large to test whether the true duration of these events are comparable. But this is tantalising and really important because it implies that T1 was not alone with its two rapid NH temperature jumps, and that T3 likely had at least one comparable rapid warming (at least in this part of the N Hemisphere) well after it started. We know from T1 that the BA transition occurred âĹij5 kyr into the termination.

Comparisons of the millennial scale shifts in North Atlantic climate have been previously examined in Pérez-Mejías et al. 2017 and Cheng et al. 2007. For example, Cheng et al. 2007 define the timing of weak and strong monsoon intervals: YD-III and BA-III. Pérez-Mejías et al. 2017 suggest that the S8.2 IRD event in the marine record triggered Heinrich Stadial-like conditions in southwestern Europe, similar to Heinrich Event 1 prior to TI and Henrich Event 11 prior to TII. Our findings support the timing of these shifts in North Atlantic climate. However, we agree with the reviewer that further comparison and discussion is needed in our manuscript. We have expanded the discussion of millennial scale events leading up to TIII in the "onset of deposition" discussion section (line 187). We also highlighted the intervals YD-III, BA-III, and Heinrich Event-like S8.2 in figure 5 and added an explanation to its associated caption.

6. There is an alternative explanation the authors should consider too: is the speleothem 18O acting like an 'on-off' switch, i.e. does it represent binary swings between (i) periods when the glacier is present above the cave (when basal meltwaters with low 18O values derived from strongly 18O-depleted glacial or stadial snowfall occurring 1000-1200 m higher than the cave itself, near the Hintertux glacier summit âĹij3500 m a.s.l.) and (ii) periods when the glacier retreats during interglacials and interstadials and exposes the cave recharge area to direct infiltration (at âĹij2300 m) of isotopically enriched rainfall and in situ snowfall? This could explain the almost square-wave form and amplitude of the speleothem isotopic series (otherwise for the

MIS7a-MIS6 transition, for example, we must consider 20 deg C or more of tempera-ture depression plus a little extra for possible changes in moisture source, given the >6 per mil decrease in speleothem 18O). This raises the question of whether the sharp increases and decreases in d18O are really a local effect of ice retreat, whose phas-ing with respect to regional warming and cooling (e.g. the rises and falls in SST in MD01-2444) is not as closely coupled as the authors think.

The reviewer proposes an interesting hypothesis. However, several lines of evidence argue against this. As outlined in Spötl et al (2008), there is strong evidence that the cave was continuously buried by the local glacier during MIS 7 (and actually most of the Pleistocene – see Spötl & Mangini, 2007 EPSL). The sampling location of SPA 121 was underneath glacier ice up to the end of the 19th century and was very close to the ice margin still at about 1920 AD (i.e. well within an interglacial).

It is incorrect that the precipitation that fed cave dripwaters fell 1000-1200 m higher. The highest summit of Hintertux glacier (Olperer) is 3476 m, whereas the glacier basin (main accumulation area) is located at about 2800-3000 m. This is only a few hundred meters above the speleothem sampling site.

Another line of evidence is that the C isotope composition remains within the limited range of host rock values over tens of thousands of years (except for some of the cold periods when we see high values, which are very likely kinetically controlled—as discussed in the responses to reviewer #1). If the area above the cave would have become deglaciated e.g. during MIS 7a, we would expect colonization of alpine veg-etation above the cave within a few decades, as shown by many recent observations. This would result in a drop in d13C values, which is not observed. The reviewer also mentioned the possibility of intermittent waxing and waning of the Hintertux glacier, thereby resulting in meltwater and sediment pulses. While this is an interesting to con-sider, there is currently no evidence in the petrography of the stalagmite nor in the d18O values for melt water pulses.

Regarding the large decreases in d18O: TII records from Spannagel show a shift of about 3.5 to 4 per mil (Spötl et al., 2002 Geology; Holzkämper et al., 2004 GRL). This amplitude is similar to the shifts observed in SPA 121 during MIS 7, with the exception of the 4.5 per mil decrease at the end of MIS 7a. Following the termination of SPA121 growth, an additional 1.5 per mil decrease is recorded in stalagmite SPA 183 – resulting in a 6 per mil decrease in d18O between MIS 7a and 6. The difference in absolute d18O values between SPA 121 and SPA 183 was previously addressed in reviewer 1's responses. In short, we interpret the cumulative 6 per mil decrease as a signal for regional cooling that was amplified by kinetic isotope fractionation effects.
* * *
[Figure]

| Marine Isotope Stage | Onset of associated climate changes in the European Alps (ka) |
|:---:|:---:|
| TIII | 242.5 ±0.3 |
| MIS 7e | 241.8 ±0.3 |
| MIS 7d | 234.3 ±0.3 |
| TIIIa | 216.8 ±0.3 |
| MIS 7c | 215.7 ±0.4 |
| MIS 7b | 211.7 ±0.4 |
| MIS 7a | 201.8 ±0.3 |
| MIS 7/6 transition | 197.1 ±0.2 |

**Fig. 1.** Table 1: The onset of regional climate changes associated with MIS 7 sub-stages and the MIS 7/6 transition, as defined by the Spannagel $\delta18O$ record (see text for detailed definitions).

---

## Author Response (AR1)

**Response to reviewers for the Wendt et al. manuscript: "Precise timing of MIS 7 sub-stages from the Austrian Alps"**

We greatly appreciate the comments and critiques made by the two anonymous reviewers. Below is a list of individual comments and questions. Our responses are in blue:

**Reviewer #1**

1. I am not very convinced of the data presented from the two new stalagmites, and I think the authors should consider the benefits of including them here.

We agree that our original manuscript lacked sufficient evidence to ensure that the two new stalagmites (SPA 146 and 183) precipitated in isotopic equilibrium with dripwaters. To address this issue, we have re-sampled 8 Hendy tests from SPA146 and 5 from SPA183 (see new supplementary figures below). The results present a more comprehensive picture of the two stalagmites. The lack of 18O and d13C covariation suggest that SPA 146 and SPA 183, like SPA 121, grew in isotopic equilibrium.

It's important to note that the stable isotope values from each stalagmite do not replicate each other perfectly. This may be due to several factors (see responses below for additional discussion). However, we argue that the isotopic trends generally agree and can be interpreted in relative terms as a proxy for past regional climate changes. To this end, we maintain that stalagmites SPA 146 and 183 provide valuable insight into the timing of and regional cooling at the onset of MIS 6.

[Figure]

New Supplementary Figure A (to replace supplementary figure 3)

**Response to reviewers for the Wendt et al. manuscript: "Precise timing of MIS 7 sub-stages from the Austrian Alps"**

[Figure]

New Supplementary Figure B (to replace supplementary figure 3)

2.- Source of precipitation. The authors indicate two main sources of precipitation in this region, which can be differentiated by the d18O isotopic values. I agree with this statement, but I consider that Atlantic sourced precipitation may not be much more negative than the Mediterranean one, depending on the moisture uptake along the longer pathway. Rainout effect is sometimes compensated by the more positive recycled moisture that is being incorporated in the way from the source to the Central Alps. It is then important to take into account the moisture recharge through the long pathway as, sometimes, the result is an enrichment derived by the effect of enriched inland moisture compared to ocean moisture. See, for example, Chakraborty et al., (2016) and Krklec and Domínguez-Villar (2014).

Our original text cited a combination of site-specific modern precipitation and dripwater calibration studies that support our interpretation of Spannagel d18O. An expanded discussion of precipitation d18O in the Austrian Alps is now provided in a new section titled "climate setting"

We agree with the reviewer that, due to rainout effects, the difference in d18O of North Atlantic vs Mediterranean sourced rainfall to our study site likely minimal. We have reworded the text emphasize that temperature is the dominate control on Spannagel d18O, with only minimal (amplifying) effects related to source.

**Response to reviewers for the Wendt et al. manuscript: "Precise timing of MIS 7 sub-stages from the Austrian Alps"**

3. Similarity with d18O monsoon records. The authors indicate several times in the discussion the high similarity with Asian monsoon records (lines 175, lines 235, etc); I think these statements should be modulated as I observe many differences in timing and pattern in Fig. 4. Both the similarities and the differences must be clearly described. For example, the time of TIIIa is completely different, also the pattern. The time of 7d as defined in Spanagel (234-216 ka) does not coincide at all with Chinese monsoon timing. Please indicate and explain potential mechanisms for those differences.

We agree that our original text lacked a full-picture explanation behind the interpreted similarities between the Chinese monsoon and Spannagel records. For example, we highlighted the similarities between variations in the Spannagel $\delta^{18}O$ and Chinese Monsoon $\delta^{18}O$ on sub-orbital timescales, but did not explain why the two locations are decoupled on other timescales (e.g. orbital) which may cause confusion to readers. We have now expanded our discussion of Spannagel $\delta^{18}O$ vs Chinese Monsoon $\delta^{18}O$ and revised Figures 3-5.

Regarding the timing of 7d: we agree with the reviewer that Fig. 4 and its related text may render confusing without a clear reiteration of age uncertainties associated with both records. For example, the period of maximum isotopic depletion of MIS7d in Spannagel (229.2 ka) appears early, but is actually within the ~1 ka age uncertainties of the Chinese Monsoon record (228.2 ka). Nevertheless, our original text lacked an emphasis on the respective age uncertainties of each record. We have reworded these and similar statements to include a comparison of age uncertainties.

The end of MIS 7d is marked by the onset of TIIIa, which according to Cheng et al. 2016 occurs in the Chinese monsoon record at 217.1 ±0.9 ka. The timing of TIIIa is therefore well within uncertainties of both records. To further emphasize this, we have expanded the TIIIa discussion section and revised Figure 4.

- Line 26. I miss one or two references here to support this statement.
We added the citation PAGES (2016) as a summary reference for MIS 7 substages.

- Line 140. Replication just happens during very short periods of time, if any, and the values and trends are not so well reproduced. I would not use those criteria for discarding kinetic effects.
We have re-sampled multiple Hendy tests for SPA146 and 183 in order to provide a more robust test for possible kinetic effects (see response to first comment). It is true that the absolute d18O and d13C values from each stalagmite do not replicate each other perfectly. This may be due to several factors (see response below). However, we argue that the isotopic trends generally agree between stalagmites, particularly in their trend towards depleted values at the onset of MIS 6. Returning to the reviewers' point, we no longer argue that replication is the main line of evidence against kinetic effects; instead, we point to the evidence provided by the new Hendy test analysis.

- Line 147. This just applies for SPA21, the other two stalagmites display more negative values. Please, explain why.
Spötl et al (2008) noted that prior calcite precipitation (PCP) likely influenced the stable isotope values of SPA121. Two lines of evidence in this study supports this hypothesis. First: SPA 121

**Response to reviewers for the Wendt et al. manuscript: "Precise timing of MIS 7 sub-stages from the Austrian Alps"**

grew very slowly relative to SPA 146 and 183, suggesting a slow drip rate (and thus, higher likelihood of PCP). Second: low d18O intervals recorded in SPA 121 coincide with high d13C (e.g. during the MIS 7a-6 transition), suggesting the influence of a kinetically-controlled process, such as PCP, during cold intervals. We thank the reviewer for raising this question and have added a more detailed explanation to the text.

- Figure 4. I would suggest adding to this figure the duration of MIS7 substages (lines or shaded squares) to really see when they start and finish, not only the "peak" indicated by the name in Fig. 4D.
Done

- Figure S3. I think these data correspond to two different laminae in every stalagmite. Please indicate it in the graph or caption.
We have replaced Figure S3 (see response to first comment)

**Response to reviewers for the Wendt et al. manuscript: "Precise timing of MIS 7 sub-stages from the Austrian Alps"**

**Reviewer #2**

In my opinion, the authors misuse the terms 'transition' and 'termination'. The point I want to make is that speleothems do not preserve terminations or other MIS transitions per se. Ocean sediments do

We agree that our interpretation and use of the term "termination" was used too liberally in our original text. As the reviewer correctly pointed out, this paper is a study of d18O changes speleothems from the central European Alps. Although evidence suggests that Spannagel d18O is highly sensitive to changes in the North Atlantic realm, speleothems (as with all terrestrial archives) cannot be used to directly infer large-scale changes in the ocean-cryosphere system. To this end, we have revised the manuscript to avoid such sweeping statements. We also expanded the first paragraph of our discussion section to emphasize that Spannagel d18O records the regional expression of climate changes associated MIS 7 sub-stages and terminations.

The authors do refer to ocean records in the ms (LR04, MD01-2444: Figure 4) but do not determine exactly how the Alpine speleothem 18O profile links to these records, apart from references to alpine warming coeval with SST increases (and the converse)… The case for a link between the cave and ocean records through the whole time interval must be better developed.

We agree that an expanded discussion of this topic is needed. North Atlantic sediments show clear evidence of SST warming and deposition of ice-rafted debris associated with the rapid transition of TIII and TIIIa (e.g. Martrat et al. 2007; Channell et al. 2012). The onset of these oceanic changes occurs within age uncertainties with an abrupt enrichment in Spannagel d18O. The close coincidence between marine and terrestrial records suggests that warming temperatures in the North Atlantic realm drove warmer winter temperatures in the central Alps during these time periods. However, is it also clear that Spannagel d18O does not remain coupled with North Atlantic SSTs throughout MIS 7, such as during MIS 7d, which suggests that there are multiple driving forces that influence winter temperatures in the central Alps. We have added a discussion of the complex link between the North Atlantic and central Alps in the first paragraph of the discussion section and added a new section called "climate setting" to provide greater context.

Regarding the interval of stable d18O values between 247 and 242 ka: to which part of the ocean record does this correspond? Is it the 'late MIS8 glacial' before the termination actually starts (it would seem so, based on the authors' claims of a short termination that starts after this isotope plateau), or is it really part of the period of ice-sheet melting associated with the termination, as implied in Cheng et al. 2016 and 2009, and Pérez-Mejías et al. 2017? If the latter, which in my opinion (based on all the evidence) is more realistic, the quoted ages and durations of T3 and potentially other transitions listed in table 1 have little meaning.

We agree - our record cannot rule out the possibility that the depleted isotope plateau between 247 and 242 ka is associated the onset of ice sheet melting associated with Termination III isotope plateau. It is therefore incorrect to define the timing and duration of TIII in the strict terms outlined in our first draft. To avoid confusion, we have eliminated table 1.

**Response to reviewers for the Wendt et al. manuscript: "Precise timing of MIS 7 sub-stages from the Austrian Alps"**

From what I can determine, it seems that the speleothem did not even capture all of T3, if you take into consideration previously published speleothem records (Cheng et al. 2009 and 2016 and Pérez-Mejías et al. 2017). It obviously captures all of 7e, the 7e/d transition and the 7d/c transition (T3a), etc. but exactly how do the boundaries of these transitions in the ocean record tie to the speleothem 18O?

If one defines the main portion of TIII as the period of maximum IRD deposition in the North Atlantic between 243-240 (Channell et al. 2012), then the period between the onset of speleothem deposition at 247.3±0.2 ka and the abrupt 3‰ increase of Spannagel d18O from 242.5 to 241.9 (±0.3) ka is well within uncertainties of the IRD deposition event. This time frame is also consistent with the abrupt changes with vegetation productivity in the Iberian Peninsula (241.6–240.7 ±1.6 ka; Pérez-Mejías et al. 2017) and Chinese Monsoon intensity (242.8–241.01 ±0.9 ka; Cheng et al. 2009, 2016). Spannagel may not cover all millennial-scale events leading up to TIII, but we disagree with the reviewer that Spannagel does not capture TIII.

We have added a section titled "climate setting" to provide further details on speleothem 18O.

In the context of the above, I would like the authors to carefully consider exactly what the abrupt speleothem 18O changes mean at this high altitude cave? For instance, are the abrupt increases examples of Bølling-Allerød-like or YD-Holocene-like events? Hard to say – age uncertainties, although small in percentage terms, are still too large to test whether the true duration of these events are comparable. But this is tantalising and really important because it implies that T1 was not alone with its two rapid NH temperature jumps, and that T3 likely had at least one comparable rapid warming (at least in this part of the N Hemisphere) well after it started. We know from T1 that the BA transition occurred ~5 kyr into the termination.

Comparisons of the millennial scale shifts in North Atlantic climate have been previously examined in Pérez-Mejías et al. 2017 and Cheng et al. 2007. For example, Cheng et al. 2007 define the timing of weak and strong monsoon intervals: YD-III and BA-III. Pérez-Mejías et al. 2017 suggest that the S8.2 IRD event in the marine record triggered Heinrich Stadial-like conditions in southwestern Europe, similar to Heinrich Event 1 prior to TI and Henrich Event 11 prior to TII. Our findings support the timing of these shifts in North Atlantic climate. However, we agree with the reviewer that further comparison and discussion is needed in our manuscript. We have expanded the discussion of millennial scale events leading up to TIII and TIIIa in discussion section. We also highlighted the intervals of Heinrich Event-like events in Fig. 4 and added an explanation to its associated caption.

There is an alternative explanation the authors should consider too: is the speleothem 18O acting like an 'on-off' switch, i.e. does it represent binary swings between (i) periods when the glacier is present above the cave (when basal meltwaters with low 18O values derived from strongly 18O-depleted glacial or stadial snowfall occurring 1000-1200 m higher than the cave itself, near the Hintertux glacier summit ~3500 m a.s.l.) and (ii) periods when the glacier retreats during interglacials and interstadials and exposes the cave recharge area to direct infiltration (at ~2300

**Response to reviewers for the Wendt et al. manuscript: "Precise timing of MIS 7 sub-stages from the Austrian Alps"**

m) of isotopically enriched rainfall and in situ snowfall? This could explain the almost square-wave form and amplitude of the speleothem isotopic series (otherwise for the MIS7a-MIS6 transition, for example, we must consider 20 deg C or more of temperature depression plus a little extra for possible changes in moisture source, given the >6 per mil decrease in speleothem 18O). This raises the question of whether the sharp increases and decreases in d18O are really a local effect of ice retreat, whose phasing with respect to regional warming and cooling (e.g. the rises and falls in SST in MD01-2444) is not as closely coupled as the authors think.

The reviewer proposes an interesting hypothesis. However, several lines of evidence argue against this. As outlined in Spötl et al (2008), there is strong evidence that the cave was continuously buried by the local glacier during MIS 7 (and actually most of the Pleistocene – see Spötl & Mangini, EPSL 2007). The sampling location of SPA 121 was underneath glacier ice as recently as the end of the 19th century and very close to the ice margin still at about 1920 AD, i.e. well within an interglacial.

Another line of evidence is that the C isotope composition remains within the limited range of host rock values over tens of thousands of years (except for some of the cold periods when we see the high, and likely kinetically controlled, values which we interpret this to be a reflection of partial freezing of the karst). If the area above the cave would have become deglaciated e.g. during MIS 7a, we would expect colonization of alpine vegetation above the cave (within a few decades as shown by many recent observations), resulting in a drop in d13C values. This is not observed.

Intermittent waxing of the glacier, as suggested by the reviewer, may also result in meltwater and sediment pulses. There is currently no evidence in the petrography of the stalagmite nor in the d18O values for melt pulses.

Finally, it is incorrect that the precipitation that fed cave dripwaters fell 1000-1200 m higher. The highest summit of Hintertux glacier (Olperer) is 3476 m, whereas the gentle glacier basin (i.e. main accumulation area) is located at about 2800-3000 m. This is only a few hundred meters above the speleothem sampling site.

Regarding the large decreases in d18O: TII records from Spannagel show a shift of about 3.5 to 4 per mil (Spötl et al., Geology 2002; Holzkämper et al. GRL 2004). This amplitude is similar to the shifts observed in SPA 121 during MIS 7, with the exception of the 4.5 per mil decrease at the end of MIS 7a. Following the end of SPA121 growth, an additional 1.5 decrease in d18O is recorded from stalagmite SPA 183 – resulting in a 6 per mil decrease in d18O between MIS 7a and 6. The difference in absolute d18O values between SPA 121 and SPA 183 was previously addressed in reviewer 1's responses. In short, we do not interpret the cumulative 6 per mil decrease to be a climatic (or temperature) drop alone – but instead a function of different kinetic components between stalagmites.

---

## Author Response (AR2)

**Response to reviewer #2 (**authors answers in blue**)**

1. The Hendy test plots are incorrect (I am going off the Supp A and B figure versions reproduced in the "author response" file): the C and O axis labels should be swapped.

Supplementary figures 4-5 have been corrected.

2. The conclusion that there is no correlation between C and O for the Hendy tests is not convincing. Many look well correlated (e.g. H1 to H3 of Supp B, and H1 to H6 in Supp A - it is hard to tell because the axis ranges on the right square plot of each plot pair are too wide, and conceal the upward trend - please reduce the range of axis values). So I would not be so dogmatic with statements about lack of isotopic equilibrium. Fortunately, the signal in these speleothems is so large, a modest amount of kinetic fractionation is not a big issue.

We have toned down our statement regarding isotopic equilibrium (line 266).

3. Some of the drill hole positions for the Hendy traverses for Supp B are either poorly aligned on the image or do not represent coeval calcite (I certainly hope it is the former). For the sake of future readers, it would be helpful to have this corrected / made more accurate, as it does not set a good example for those speleothemists new to the game.

Supplementary figures 4-5 have been corrected.

4. I remain adamant that the speleothem does not capture all of TIII because the older H-event (S8.2) is part of the termination, as correctly inferred in both Cheng et al. 2009 and Pérez-Mejías et al. 2017. The speleothem clearly started growing after S8.2, between S8.2 and S81, so it captures most of TIII but not all of it.

We have added the description "later part of TIII" to underscore this point throughout the abstract and discussion.

5. The time series from Columbu et al. (2019) is actually from Crovassa Azzurra Cave in Sardinia (Figure 5).

Figure 1 & 5 have been corrected.

6. Re contribution from the summit of Hintertux glacier: based on the oblique-angle image of Figure 1, it seems that during glacier advances, ice formed at or near the summit of the massif could flow under gravity over the surface above the cave and its meltwaters could infiltrate the cave - so meltwater from a higher altitude could reach the cave, unless my sense of 3D is letting me down.

Reviewer #2 is correct that meltwater from higher altitudes can infiltrate into cave. However, it is incorrect that the precipitation fell several km higher (as the reviewer argued in the first

revisions). The highest summit of Hintertux glacier (Olperer) is 3476 m, whereas the gentle glacier basin (i.e. main accumulation area) is located at about 2800-3000 m. This is only a few hundred meters above the speleothem sampling site. This also holds true for MIS 7, when the glacier tongue buried the cave.

7. Line 49: should be TIIIa, not TIIa.

Corrected.

8. Line 161: 'long and short timescales': please be more specific, e.g. orbital vs millennial (or centennial).

Clarified.

9. Line 234: To the best of my knowledge, the process of PCP cannot directly affect d18O unless evaporative enrichment occurs. Please provide a reference supporting this point. I would have thought differences in the recharge altitude of individual drip points (not out of the question for a deep cave set in mountainous terrain) could comfortably explain small d18O offsets.

We have removed this argument.

10. Throughout the Discussion, references to your figures are needed.

Corrected.

---

## Author Response (AR3)

Response to editor: thank you very much for suggesting these final details. Below are our answers in blue.

- Define "ka" in the abstract
Done

- Decide if you use brackets or not each time you give uncertainty for the dating but do the same for each occurrence
Thanks for catching this inconsistency. We quote U-Th uncertainties using the following rule: use parentheses for multiple ages but no parentheses for a single age. For example: 234–235 (±0.2) ka versus 234 ±0.2 ka. Our earlier draft was not consistent with the rule. We have made these corrections.

In addition, we changed "-" for "–" when necessary.

- Avoid red writing on Figure 1 (difficult to read)
Corrected

- Figure 2: better use ka than years BP
Corrected

- Figure 3: increase font size of for "Precession", "Obliquity", "insolation"
Corrected